# Energy Drink Consumption among Adolescents Attending Schools in Lubumbashi, Democratic Republic of Congo

**DOI:** 10.3390/ijerph18147617

**Published:** 2021-07-17

**Authors:** Trésor Carsi Kuhangana, Taty Muta Musambo, Joseph Pyana Kitenge, Tony Kayembe-Kitenge, Arlène Kazadi Ngoy, Paul Musa Obadia, Célestin Banza Lubaba Nkulu, Angélique Kamugisha, Eric Deconinck, Benoit Nemery, Joris Van Loco

**Affiliations:** 1Department of Public Health, Faculty of Medicine and Public health, University of Kolwezi, Kolwezi 07301, Democratic Republic of Congo; 2Unit of Toxicology and Environment, School of Public Health, University of Lubumbashi, Lubumbashi 07601, Democratic Republic of the Congo; tatymuta17@gmail.com (T.M.M.); docteurpyana@gmail.com (J.P.K.); tonykayemb@gmail.com (T.K.-K.); arlene.kazadingoy@gmail.com (A.K.N.); musa.p.obadia@gmail.com (P.M.O.); clubabankulu2017@gmail.com (C.B.L.N.); 3Ministry of Public Health, Haut-Katanga Provincial Inspection of Health, Lubumbashi 07601, Democratic Republic of the Congo; 4Occupational and Environmental Health Unit, Department of Public Health, University of Lubumbashi, Lubumbashi 07601, Democratic Republic of the Congo; 5Ministry of Public Health, Haut-Katanga Provincial Division of Health, Lubumbashi 07601, Democratic Republic of the Congo; 6Centre for Environment and Health, Department of Public Health and Primary Care, KU Leuven, 3000 Leuven, Belgium; 7Physical and Chemical Health Risks, Sciensano, 1050 Brussels, Belgium; angelique.kamugisha@sciensano.be (A.K.); Eric.Deconinck@sciensano.be (E.D.); Joris.VanLoco@sciensano.be (J.V.L.); 8Clinical and Experimental Endocrinology, Biomedical Sciences Group, KU Leuven, 3000 Leuven, Belgium

**Keywords:** adolescent pupils, energy drink consumption, alcohol, caffeine

## Abstract

Background: The consumption of energy drinks (EDs) is increasing in the general population, but little is known about the consumption of EDs among pupils in Africa. This study was designed to assess the consumption of EDs among pupils between 10 and 17 years of age and to assess average caffeine concentrations contained in EDs sold in Lubumbashi. Methods: We conducted a cross-sectional survey in five schools using a standardised questionnaire taken face-to-face. Samples of locally purchased EDs were analysed by High Performance Liquid Chromatography with Ultra-Violet spectrometry (HPLC-UV). Results: Of 338 pupils (54% girls), 63% reported having consumed at least one ED in the last week and 34% drank at least one ED a day. The cheapest ED was the most widely consumed. Among pupils having consumed at least one ED in the last week, 79% reported consuming it for refreshment and 15% to get energy. For those who reported not consuming EDs, 40% reported that their parents or teachers forbade them to drink EDs. Some (14%) teenagers, mainly boys, mixed ED with alcohol. The concentrations of caffeine measured in twelve brands of EDs ranged from 7.6 to 29.4 mg/100 mL (median 23.3), giving caffeine contents of 37.5 to 160 mg (median 90 mg) per can or bottle. The estimated daily intake of caffeine through EDs was between 51.3 mg and 441.3 mg among those consuming EDs regularly. Conclusion: Our study convincingly demonstrates that caffeine-containing EDs are not only consumed by youngsters living in affluent societies. We documented widespread regular consumption of EDs among (pre-)adolescent schoolchildren living in Lubumbashi, a large city of the Democratic Republic of Congo (DRC). In view of the global market expansion of caffeinated EDs, it is reasonable to suspect that similar surveys in other urban areas of sub-Saharan Africa would yield similar findings. Pricing and advertising regulations and education on EDs are necessary to limit the regular consumption of EDs among adolescents.

## 1. Introduction

In recent years, the consumption of energy drinks (EDs) has considerably increased worldwide through tactical marketing and advertising strategies [1]. Occasional or moderate consumption of EDs seems to entail a low risk for healthy people; nevertheless, excessive consumption of EDs, especially together with alcohol and/or illicit drugs, may represent a health hazard for young people or individuals with cardiac, neurologic or kidney disorders [2].

The main substance associated with adverse effects in EDs is caffeine [3,4]. Excessive and regular caffeine intake may lead to a state of chronic intoxication, characterized by headaches, palpitations, agitation, irritability, restlessness, muscular jolts, tremors and gastrointestinal discomfort. Children and teenagers are more likely to experience these undesirable health effects when consuming caffeine [5]. The possible role of ingredients other than caffeine (e.g., taurine) in the effects of EDs remains unclear [6]. Moreover, frequent consumption of EDs has been associated with unhealthy behaviours, such as use of alcohol, tobacco, illicit drugs, sexual risk-taking and violence among adolescents and young adults [7,8,9,10,11,12]. Mixing EDs with alcohol has been shown to be common among high school seniors [13].

A survey conducted in the United States of America (USA) found that 51% of 496 college students surveyed reported drinking more than one ED each month [14]. According to the Québec Health Survey of High School Students, 17% of high school students drank EDs at least twice a month [15] and in 2017, 16% reported ever consuming more than two EDs in a day [16]. In South Africa, an association was found between advertising exposure and ED consumption [1].

However, little is known about the consumption of EDs elsewhere in Africa, although these drinks are widely advertised and sold, including in the Democratic Republic of Congo (DRC), a low-income country with a per capita GDP of USD 581 in 2019 [17] and the second largest and fourth most populated country of Africa. In recent years, Lubumbashi, the DRC’s second largest city and the capital of the highly industrial province of Haut-Katanga, has witnessed a substantial growth in the local production of EDs, which are sold in addition to those imported from neighbouring country Zambia. This is accompanied by a lot of attractive advertising on billboards and television.

This study was designed to assess the consumption of EDs among adolescent pupils between 10 and 17 years of age in Lubumbashi and to assess the caffeine concentrations in EDs sold in the country.

## 2. Materials and Methods

### 2.1. Participants and Survey Procedures

Data for this cross-sectional study were collected from 30 May to 7 June 2019 in five schools situated in three neighbourhoods of Lubumbashi: Complexe Scolaire les Amis de Daniel, Complexe Scolaire PP UNILU and Complexe Scolaire Rahaman in Kasapa Quarter, Complexe Scolaire Cité Divine in Gambela Quarter and Complexe Scolaire Les Missionnaires in Kamisepe Quarter. These private schools were selected based on the standards of school buildings (CIRCULAIRE N°MINEPSP/CABMIN/010/2012 DU 11/10/2012 RELATIVE AUX DIRECTIVES SUR LES NORMES DES CONSTRUCTIONS SCOLAIRES), including a fenced school with at least 6 primary and 6 secondary classrooms. The classes included were those from 6th primary to 6th secondary. Three to five days before the surveys, the headmasters and/or their collaborators in charge of discipline informed parents about the survey through class diaries; the pupils received information on the day of the survey. Participation was voluntary and without financial implication. Confidentiality was guaranteed.

Although the regional local language is Swahili, the questionnaire was elaborated and administered in French without translation because French is the educational language in DRC. The questions were based on the brand of EDs consumed, the reason for consuming EDs or not, the frequency of EDs consumption and its mixing with alcohol.

On the day of the survey, volunteering pupils in turn left their classroom to join one of two trained investigators sitting in discrete places in the schoolyard to take the questionnaire. The questions were administered face-to-face and in French (the teaching language); the questions related to demography and ED consumption including the use of alcoholic beverages; completing the questionnaire took approximately 5 min per participant. To avoid mutually influencing each other, pupils had to leave the schoolyard after having participated in the survey. Some older pupils preferred to keep their age secret because they were ashamed; only teachers or headmasters have access to the administrative files of pupils, and they are not allowed to disclose the content without the pupils’ consent. Some older pupils possibly participated in the survey to prove to their colleagues that they were normal school age. To respect privacy, ages were at our disposal at the end of the survey.

At the University of Lubumbashi and the University of Kolwezi, studies involving clinical examination, invasive procedures (such as taking biological samples) or therapeutic interventions must receive prior approval from the University’s medical ethics committee. However, surveys conducted among persons who are not patients and are limited to answering an anonymous questionnaire do not require approval by the medical ethics committee. This study was approved by the local school authorities, parents received prior information, and the surveys were conducted in strict compliance with research standards, including voluntary participation and respect for privacy and confidentiality for each participant. The study was carried out in compliance with Congolese law with the collaboration of the Ministry of Public Health through the Programme National de Lutte Contre les Toxicomanies et les Substances Toxiques (National Program for the Fight against Drug Addiction and Toxic Substances (PNLCT)).

### 2.2. Patient and Public Involvement Statement

Study participants were not directly involved in organizing the survey, but school officials and teachers were involved. In each potentially participating school, a meeting was held with them before starting the survey in order to present the study and its importance. After having a clear idea of what would be done, school officials requested permission for student participation from each parent, before contacting the investigator team. Additionally, in each school, the investigation began with a short speech from the head of the school or his delegate, reminding pupils of the merits of the study and their free participation, while assuring them that a refusal of participating would have no negative or positive impact on their school evaluation. In each school visited, some pupils’ teachers were made available to us for carrying out the survey.

It was agreed with school officials and teachers that a copy of the study results would be presented to them, so as to include it in their teaching and awareness messages to pupils. In addition, the authors intend to present the results of the study at youth awareness days in collaboration with other schools via the Ministry of Primary, Secondary and Technical Education, partners working in the health promotion and education, and the Ministry of Public Health through the PNLCT. At the higher education level, the results of the study will also be presented in an academic conference.

### 2.3. Energy Drink Collection and Chemical analysis of Caffeine

All available types of EDs (cans and bottles) were bought in various supermarkets of Lubumbashi in December 2018. Information concerning the ingredients was obtained from the labels on the bottles or cans, but we did not address the issue of nutritional value because the focus of our study was on caffeine content. Samples of 4 mL were transferred into cryovials and stored at −20 °C in the laboratory of the Unit of Toxicology and Environment of the University of Lubumbashi. The samples were transferred to Belgium in cool boxes, stored frozen and then analysed for the concentration of caffeine by High Performance Liquid Chromatography with Ultra-Violet spectrometry (HPLC-UV) in a governmental ISO17025-certified laboratory specialized in the analysis of medicines and health products (Sciensano, Brussels, Belgium).

### 2.4. Statistical Analysis

Statistical analysis was performed with SPSS (version 20.0; SPSS Inc., IBM, Chicago, IL, USA). The comparisons of proportions were calculated using Pearson Chi-Square test. The threshold level for significance was set at *p* less than 0.05.

## 3. Results

A total of 422 pupils from 6th primary grade to 6th secondary grade (48 to 155 per school) participated in the survey. Their ages ranged from 10 to 23 years (it is not unusual in the DRC to have students older than 18 years attending secondary school), but we excluded 84 participants older than 17 years because we targeted early pre-adolescents (10–13 years) and middle adolescents (14–17 years) [18]. We also excluded one participant who refused to continue replying to the questionnaires.

Thus, our results are based on data obtained from a total of 338 pupils aged 10 to 17 years. For the analysis, we divided participants into 147 pupils (57.8% of girls) aged 10 to 13 years (mean 12.1, SD 0.9) and 191 pupils (51.3% of girls) aged 14 to 17 years (mean 15.5, SD 1.2).

Table 1 shows information about the 12 types of bottles or cans of EDs sold in Lubumbashi in December 2018. Three brands were produced in the DRC, five came from African countries (three from neighbouring Zambia) and four were labelled as originating from Europe or Korea. Unit volumes ranged from 250 to 500 mL and unit prices ranged from 500 to 2500 Congolese Francs (median 1500), i.e., approximately USD 0.30 to 1.50 at the time of the survey. The cheapest, locally produced brand, Bora Boom, did not provide nutritional information, but the caffeine concentrations indicated on the labels of the 11 other brands ranged from 15 to 32 mg/100 mL (mean 28.2, SD 5.3), giving a total nominal content of 37.5 to 160 mg caffeine (mean 98.7, SD 35.5) per can or bottle. The measured caffeine concentrations ranged from 7.6 to 29.4 mg/100 mL (mean 22.4, SD 6.6), giving caffeine contents of 19 to 147 mg (mean 77, SD 32) per can or bottle. In other words, in the 11 brands where this could be assessed, the measured caffeine concentrations proved to be consistently (*p* < 0.001 by Student’s paired test) lower than those mentioned on the label by a mean of 5.7 mg/100 mL (range −0.6 to −11.4 mg/100 mL), thus giving a 20.5 mg lower mean caffeine content per bottle/can (range −3 to −57 mg).

The responses to the questionnaire are summarized in Table 2. Among the 338 respondents, 126 (37%) declared not to consume EDs or to consume less than one drink per week; the proportion of non-consumers was higher in the younger age category (44%) than in the older age category (32%). The reasons given for not consuming EDs varied significantly by age category: prohibition by parents or teachers was put forward more frequently by younger pupils (58%) than by older pupils (20%), whereas older non-consumers of EDs more frequently did not (or would not) give reasons than younger non-consumers (54% vs. 38%, respectively); older self-declared non-drinkers evoked “health complications” (palpitations and absence of sleeping) more frequently than younger non-drinkers (26% vs. 3%, respectively).

Among the 212 respondents who reported consuming EDs at least once a week, most (79%) declared to drink just for refreshment, both among the younger pupils (77%) and the older pupils (81%), and 15% declared to do so “to have energy” (11% among younger pupils and 17% among older pupils). Among those who consumed at least one ED per week, 115 (54%) consumed at least one drink per day; sixteen and five of these daily drinkers declared consuming two and three EDs per day, respectively; most (18/21) of these “heavy” consumers were older than 13 years. Among the 212 regular consumers of EDs, 30 (14%) said that they mixed these drinks with alcohol, again the majority (27/30) being older than 13 years. The most widely consumed brand of ED was Bora Boom (77%), followed by Kung Fu (15%), with little differences between the age categories.

No significant differences were found between the proportions of girls and boys consuming EDs (Table 3). However, in regard to the reported reasons for consuming EDs, female pupils responded consuming EDs for refreshment (106/116, 91%) more frequently than male pupils (62/96, 65%), this being the case in both age categories; conversely, fewer girls said that they consumed EDs to get energy (4/116, 3%) compared to boys (48/96, 50%). Among 14–17 year-old regular consumers, males more frequently reported mixing EDs with alcohol (23/57, 40%) than females (4/72, 6%). 

Photographs taken in Lubumbashi are shown in the Supplement. Appendix A shows photographs of three popular EDs (Bora Boom, Kung Fu and XXL); Appendix A shows various adverts for EDs; Appendix A shows photographs of a mother giving a young child an ED to drink and schoolchildren with EDs.

## 4. Discussion

Our study revealed that a large proportion of (pre-)adolescent pupils of five schools in Lubumbashi regularly consumed EDs, that the prevalence of regular ED consumption increased with age and that some teenagers mixed ED with alcohol. Laboratory measurements allowed us to determine the caffeine content of EDs available in the country and to estimate a caffeine intake of 51 mg to 441 mg per day among those consuming EDs. The most widely consumed brand of ED was the cheapest, locally produced drink and it did not provide nutritional information.

We acknowledge upfront the methodological limitations of our survey. Our participants consisted of an opportunistic sample of pupils from five schools, who volunteered to respond to a brief questionnaire, and we did not evaluate the socioeconomic characteristics of our respondents. Such opportunistic sampling may lead to biases that could affect the generalizability of the obtained results. A first possible bias concerns the participation by the pupils. The participation rates in the different schools were unknown because we were unable to obtain exact numbers of pupils per class or per school on the survey days since the headmasters were reluctant to provide these figures, arguing that our survey was not an inspection by the Ministry of Education. Based on an estimated total target of 1750 pupils (7 classes of 50 pupils in each of 5 schools) and 442 pupils interviewed, the percentage of respondents was approximately 24% (14% to 44% per school), although, it must be noted that 84 respondents above 17 years of age were excluded from the analysis. We do not know if ED consumers were less or more likely to volunteer for the survey, but we think this probably did not play a major role because some pupils who had initially replied to the school discipline manager that they were not interested in the survey, wanted to participate after the survey was closed. However, we could no longer include them since they had already been in contact with those who had participated. In sum, we have no reason to believe the participants did not reflect the pupils in each of the included schools.

However, a more critical risk of bias relates to the representativity of the surveyed schools. Because the inclusion of public schools would have required long administrative procedures, we decided to approach private schools situated in representative neighbourhoods of the urban area of Lubumbashi and we thus succeeded in including one of the two large Muslim schools existing in Lubumbashi, two Christian schools and two schools without religious tendencies. Although socioeconomic factors were not formally assessed, the selected schools were not the most expensive private schools in town and, hence, our participants were probably neither among the richest nor the poorest children of the population. So, although we cannot exclude participation or recruitment biases, it seems reasonable to consider that the findings of our survey reflect the prevailing pattern of ED consumption among school-going adolescents in Lubumbashi.

To our knowledge, this study is the first investigation of ED consumption among adolescent pupils in the DRC. Although the survey was conducted in Lubumbashi only, we have no reason to believe that the situation would be very different in other urban areas of the country.

In our survey of adolescent pupils, the majority (62.7%) reported consuming at least one ED per week, with half of them (54.3%) consuming at least one ED per day. These figures appear higher than those observed in several studies from Asia, the USA and Europe, which showed a prevalence of adolescents reporting regular ED consumption between 20% and 55% [6,19,20,21]. The high prevalence of ED consumption among pupils attending secondary schools in Lubumbashi is compatible with a study conducted among students at the University of Lubumbashi, which showed that 97% of respondents consumed EDs, with 59.5% being regular consumers and 52.2% reporting drinking ED several times per day [22]. In a study of 375 students (aged 16–18 years) in Israel, females reported more ED consumption than males [20]. Here, we found no significant gender differences in the prevalence of regular consumption of EDs. However, the reasons for consuming EDs varied according to gender, with girls more likely to report drinking EDs for refreshment, as opposed to boys who were more likely to invoke getting energy as the reason for consuming EDs. The prevalence of (reported) mixing EDs with alcohol was also much higher among (older) males than females. The latter is in accordance with a study conducted in the USA [13,23], where it was 10% in college students aged 18 years [23]. A recent study showed that consumption of EDs can be associated with substance use such as cigarette smoking, binge drinking and opioid use among middle and high school students [24]. In young people, the primary side effects of consuming EDs include tachycardia, insomnia and tremors [25]. Mixing EDs with alcohol was associated with certain sociodemographic characteristics, academic and social factors, with other substance use and with a greater likelihood of alcohol-related unsafe driving among high school students [26]. Two studies conducted in Slovakia and Trinidad and Tobago showed that young adolescents consuming both alcohol and energy drinks were at higher risk of negative behavioural outcomes than their peers who drank only alcohol or energy drinks or were non-consumers [27,28]. Megan et al. found that consuming alcohol and EDs together led to students drinking more alcohol and, hence, reaching higher estimated peak blood alcohol levels than after drinking only alcohol [25], probably because caffeine attenuates alcohol’s sedative effects [29]. It remains to be established whether similar factors play a role in other settings.

In the current study, the reason for no consumption of ED reported by the majority of young adolescents was related to a ban, either by parents or teachers. This could mean that rule setting for any educator who cares for children and adolescents could reduce the consumption of soft drinks and EDs. In a study from Slovakia, a lack of parental rule setting on eating was strongly associated with frequent soft drink and energy drink consumption [30]. A recent study conducted in Sweden among adolescents, showed that consumption of EDs was more common among girls not living with both parents and both boys and girls who reported low levels of parental and teacher support were more likely to consume EDs [31].

The influence of commercial factors and advertising on the responses given by the pupils about their consumption of EDs, was not formally evaluated. Advertising has a great impact on the consumption of ED, as shown by the significant association between ED consumption and exposure to television advertising in South Africa [1]. A study conducted in Canada showed cars/vehicles with energy drink branding and ads on TV have an impact on the ED consumption among adolescents [32]. The photographs of billboards in Lubumbashi illustrate how adverts focus on the “energy” and “strength” provided by EDs (Appendix A). In the DRC, EDs are not only sold in shops or supermarkets but also on markets and by street vendors who may take seat in front of schools. In the UK, a study exploring children and young people’s attitudes and perceptions in relation to EDs, identified the relatively low price of EDs, their widespread availability, gendered branding and marketing as key factors of ED consumption [33]. Not surprisingly, the ED that was most frequently cited/consumed by the respondents was the cheapest product, namely the locally produced Bora Boom—in Swahili, “bora” means “better”—costing CDF 500 (USD 0.30), which is similar to the price of many soft drinks. The majority (79.2%; 91% among girls, 65% among boys) of pupils consuming EDs declared doing so for refreshment, suggesting that they considered EDs as just another type of soft drink and/or that they were not aware that EDs contain a stimulant. Of note, the presence of caffeine was not mentioned for the most popular brand, Bora Boom.

A study conducted in Nigeria estimated concentrations of caffeine ranged from 47.56 to 58.31 mg/L in energy drink samples [34]. This concentration range is low compared to the current study. Studies conducted in Saudi Arabia [35] and in Egypt [36] found the amount of caffeine in some energy drinks to be higher than that indicated on the label. In contrast, in our study, as in a study in Spain [37], the measured concentrations of caffeine (7.6 to 29.4 mg/100 mL) were below the ones mentioned on the label in all tested EDs. Nevertheless, in nine of the twelve brands of ED analysed in the present study, the caffeine concentrations exceeded the European Standard of caffeine concentration in ED (150 mg/kg of product) [38].

In our survey, almost one out of ten pupils aged 14–17 years (18/191) reported drinking two or even three cans/bottles of ED per day, corresponding to caffeine amounts of 128.2–193.3 mg (Bora Boom), 284.2–441.3 mg (Kung Fu) and 102.6–153.9 mg (XXL). The lower figures are still below the maximum recommended daily caffeine intake of 3 mg/kg/day [39] or 2.5 mg/kg/day [40] for children and adolescents, recently confirmed as “unlikely to be associated with adverse effects” in a recent systematic review [41]. However, the upper figures are clearly higher than the recommended intake and, hence, such consumption is likely to lead to a variety of symptoms such as nausea, vomiting, palpitations and sleep disturbance. That systematic review [41] concluded that scientific data were insufficient to evaluate with confidence the effect of caffeine dose on behaviour in adolescent populations. Of note, 14% of our respondents—mainly 14–17 year-old boys—reported mixing EDs with alcoholic beverages, a worrying practice that seems to have been studied mainly in adults, but not among teenagers [41].

In the present study, we did not formally address the nutritional energy provided by EDs. The energetic value provided by Bora Boom remains unknown because, as pointed out above, it had no nutritional information on the label. For Kung Fu, and XXL, the labels indicated 229.45 and 169.95 kilocalories per bottle, respectively. Thus, based on the reported number of units drunk per day, adolescent pupils consumed around 229 to 688 kcal (Kung Fu) or 170 to 510 kcal (XXL) per day.

## 5. Conclusions

Our study convincingly demonstrates that caffeine-containing EDs are not only consumed by youngsters living in affluent societies. We documented widespread, regular consumption of EDs among (pre-)adolescent schoolchildren living in Lubumbashi, a large city of the DRC. In view of the global market expansion of caffeinated energy drinks, it is reasonable to suspect that similar surveys in other urban areas of sub-Saharan Africa would yield similar findings.

In several countries, the presence and amount of caffeine must be mentioned on the label of EDs, with warnings such as “Not recommended for children or pregnant or breast-feeding women”, “High caffeine content”, “Do not consume more than (X) container(s) daily” and “Do not mix with alcohol” [40,42]. These recommendations should be followed in the DRC and other low-income countries.

The consumption of EDs should attract our attention because excessive ED consumption is associated with negative health and lifestyle outcomes and risk behaviours, although the evidence for the latter effects is not fully established [41]. The combined use of energy drinks and alcohol represents an emerging threat to public health [13]. As adolescent students constitute a nation’s future elite, compromising their physical, mental or behavioural health could affect the sustainable development of a country. Parents and educators should be aware of the signs and possible consequences of excessive ED consumption. Above all, legislation enforcement and advocacy on prices and advertising of EDs are warranted to limit the exposure of children to potentially dangerous habits such as regularly consuming EDs.

## Figures and Tables

**Table 1 ijerph-18-07617-t001:** Brands of energy drinks sold in Lubumbashi (DR Congo) in December 2018 and their caffeine concentrations on the label and as measured.

Energy Drink Brands	Manufacture Country	Container	Container Volume (mL)	Price (FC)	Price (USD)	Price /100 mL (FC)	Caffeine on Label (mg/100 mL)	Caffeine Measured (mg/100 mL)	Difference Label vs Lab (mg/100 mL)	Total Caffeine in Can/Bottle (Label) (mg)	Total Caffeine in Can/Bottle (Lab Result) (mg)	Difference Label vs Lab (mg)
Bora Boom *	DR Congo	Bottle	300	500	0.31	167	no label	21.4		no label	64.2	
Spark *	DR Congo	Bottle	350	1200	0.73	343	30	27.5	−2.5	105.0	96.3	−8.75
XXL *	DR Congo	Bottle	330	1000	0.61	303	21.2	15.6	−5.6	70.0	51.5	−18.48
Big Kick	Zambia	Can	500	1500	0.91	300	28	16.6	−11.4	140.0	83.0	−57
Kung Fu *	Zambia	Can	500	1500	0.91	300	30	29.4	−0.6	150.0	147.0	−3
Volcano *	Zambia	Can	500	1000	0.61	200	32	22.8	−9.2	160.0	114.0	−46
Azam	Tanzania	Bottle	300	1000	0.61	333	30	20.5	−9.5	90.0	61.5	−28.5
Dragon (red) *	South Africa	Can	330	1500	0.91	455	30	23.8	−6.2	99.0	78.5	−20.46
Bebida Energetica Light	Spain, Portugal	Can	250	1500	0.91	600	15	7.6	−7.4	37.5	19.0	−18.5
Atlas	Holland	Can	250	1500	0.91	600	31.5	26.8	−4.7	78.8	67.0	−11.75
Red Bull *	Austria	Can	250	2500	1.51	1000	32	28.2	−3.8	80.0	70.5	−9.5
Wake Up	Korea	Can	250	1500	0.91	600	30	28.6	−1.4	75.0	71.5	−3.5
mean			342.5	1350	0.82	433	28.2	22.4	−5.7	98.7	77.0	−20.5
sd			101.2	483	0.29	234	5.3	6.6	3.5	37.5	32.0	17.3

(*): brand reported consumed by the participants.

**Table 2 ijerph-18-07617-t002:** Reported consumption of energy drinks (EDs) among pupils participating in a questionnaire survey carried out from 30 May to 7 June 2019 in five schools in Lubumbashi, DR Congo.

	Total	10–13 y	14–17 y	*p*
**At least one ED per week**	*N* = 338	*n* = 147	*n* = 191	
No	126 (37.3)	65 (44.2)	61 (31.9)	<0.05
Yes	212 (62.7)	82 (55.8)	130 (68.1)	
**No consumption of ED**	*n* = 126	*n* = 65	*n* = 61	
Reason for not consuming ED				
No reason or refusal to answer	58 (46.0)	25 (38.5)	33 (54.1)	<0.0001
Ban by parents or by teachers	50 (39.7)	38 (58.5)	12 (19.7)	
Health complications	18 (14.3)	2 (3.0)	16 (26.2)	
**Regular consumption of ED**	*n* = 212	*n* = 82	*n* = 130	
Reason for consuming ED				
No reason or refusal to answer	13 (6.1)	10 (12.2)	3 (2.3)	<0.005
For refreshment	168 (79.3)	63 (76.8)	105 (80.8)	
For energy	31 (14.6)	9 (11.0)	22 (16.9)	
**At least one ED per day**	115 (54.2)	41 (35.7)	74 (64.3)	
One ED per day	94 (81.7)	38 (92.7)	56 (75.7)	<0.05
Two ED per day	16 (13.9)	3 (3.3)	13 (17.5)	
Three ED per day	5 (4.4)	0 (0.0)	5 (6.8)	
Mixing ED with alcohol				
No	182 (85.8)	79 (96.3)	103 (79.2)	<0.0005
Yes	30 (14.2)	3 (3.7)	27 (20.8)	
Brand of ED consumed (*n* = 212)				
Bora Boom	164 (77.4)	68 (82.9)	96 (73.8)	
Kung Fu	31 (14.6)	7 (8.5)	24 (18.5)	
XXL	6 (2.8)	1 (1.2)	5 (3.8)	
Other	11 (5.2)	6 (7.3)	5 (3.8)	

Data are numbers of respondents (percentages).

**Table 3 ijerph-18-07617-t003:** Reported consumption of energy drinks (EDs) among female and male pupils participating in a questionnaire survey carried out from 30 May to 7 June 2019 in five schools in Lubumbashi, DR Congo.

		Females	Males	*p*
At least one ED per week		*n* = 183	*n* = 155	
No		67 (36.6)	59 (38.0)	0.435
Yes		116 (63.4)	96 (61.9)	
Reason for not consuming ED		*n* = 67	*n* = 59	
10–13 y	No reason or refusal to answer	17 (41.5)	8 (33.3)	0.774
	Ban by parents or by teachers	23 (56.1)	15 (62.5)	
	Health complications	1 (2.4)	1 (4.2)	
14–17 y	No reason or refusal to answer	15 (57.7)	18 (51.4)	0.370
	Ban by parents or by teachers	3 (11.5)	9 (27.7)	
	Health complications	8 (30.8)	8 (22.9)	
Reason for consuming ED		*n* = 116	*n* = 96	
10–13 y	No reason or refusal to answer	4 (9.1)	6 (15.8)	<0.005
	For refreshment	40 (90.9)	23 (60.5)	
	For energy	0 (0.0)	9 (23.7)	
14–17 y	No reason or refusal to answer	2 (2.8)	1 (1.7)	<0.001
	For refreshment	66 (91.7)	39 (67.2)	
	For energy	4 (5.6)	18 (31.0)	
Mixing ED with alcohol				
10–13 y	No	43 (97.7)	36 (94.7)	0.472
	Yes	1 (2.3)	2 (5.3)	
14–17 y	No	68 (94.4)	35 (60.3)	<0.00001
	Yes	4 (5.6)	23 (39.7)	

Data are numbers of respondents (percentages).

## Data Availability

The data that support the findings of this study are available on request from the corresponding authors.

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
