# Peer review of "Energy Drink Consumption among Adolescents Attending Schools in Lubumbashi, Democratic Republic of Congo"

_ijerph, 2021, doi:10.3390/ijerph18147617_

Round 1

Reviewer 1 Report

The article "Energy drink consumption among adolescents attending schools in Lubumbashi, Democratic Republic of Congo" presents an interesting and generally well-written study that requires some improvement to be published in IJERPH:
- The discussion of the results should improve when compared with similar studies carried out in other countries.
- Please include information on the calories in beverages and include this aspect in the evaluation of these products, since there are some countries that have included seal legislation that allows consumers to be alerted.

Author Response

REPLY TO COMMENTS

Reviewer 1

The article "Energy drink consumption among adolescents attending schools in Lubumbashi, Democratic Republic of Congo" presents an interesting and generally well-written study that requires some improvement to be published in IJERPH:

Response: thanks for the positive comments

Discussion

- The discussion of the results should improve when compared with similar studies carried out in other countries.

Response: Thank you for this suggestion; we have added comparisons with similar studies, including the following recent ones:

  • Svensson, Å.; Warne, M.; Gillander Gådin,. K. Longitudinal Associations Between Energy Drink Consumption, Health, and Norm-Breaking Behavior Among Swedish Adolescents. Front. Public Health 2021, 9:597613. doi: 10.3389/fpubh.2021.597613
  • Visram, S; Crossley, SJ; Cheetham, M; Lake, A. Children and young people's perceptions of energy drinks: A qualitative study. PLoS ONE 2017, 12(11): e0188668. doi.org/10.1371/journal.pone.0188668

- Please include information on the calories in beverages and include this aspect in the evaluation of these products, since there are some countries that have included seal legislation that allows consumers to be alerted.

Response: Thank you for this relevant suggestion, but we did not address this issue since the focus of our study was on the presence of stimulants, i.e., caffeine, in the energy drinks. This point has been made in the revised manuscript.

Reviewer 2 Report

Paper review “Energy drink consumption among adolescents attending schools in Lubumbashi, Democratic Republic of Congo”.

The research project is in the field of the public health and habitual behaviors. It is an interesting paper and presents a good reading.

Concerns raised include:

Abstract

Lines 41-42, reconsider your statement ‘In view of the global market expansion of caffeinated Eds, it is reasonable to suspect that similar surveys in other urban areas of Sub-Saharan Africa would yield similar findings’, make that to be specific. In addition, it is good to include your recommendations.

Implications and contributions

This paragraph can be moved to the end of the paper.

Materials and Methods

For research involving human participants, ethics approval and consent form need to be obtained prior to data collection. Participant information sheet is also needed.

A brief introduction of the questionnaire is necessary.

Results

Lines 154-155, that could be avoided at the beginning of the study if the participant information sheet has clearly stated the criteria of participation.

Other comments

Please read the paper carefully with regard to correct English

Author Response

REPLY TO COMMENTS

Reviewer 2

Comments and Suggestions for Authors

Paper review “Energy drink consumption among adolescents attending schools in Lubumbashi, Democratic Republic of Congo”. The research project is in the field of the public health and habitual behaviors. It is an interesting paper and presents a good reading.

Response: thanks for the positive comments

Concerns raised include:

Abstract

Lines 41-42, reconsider your statement ‘In view of the global market expansion of caffeinated Eds, it is reasonable to suspect that similar surveys in other urban areas of Sub-Saharan Africa would yield similar findings’, make that to be specific. In addition, it is good to include your recommendations.

Response: Thank you for this suggestion for the abstract, a sentence of recommendation was added in the abstract.

Implications and contributions

This paragraph can be moved to the end of the paper.

Response: Thank you again for this suggestion, which we have followed in the revised manuscript.

Materials and Methods

For research involving human participants, ethics approval and consent form need to be obtained prior to data collection. Participant information sheet is also needed.

Response: As we explained in the methodology section, at the University of Lubumbashi, studies involving clinical examination, invasive procedures (such as taking biological samples) or therapeutic interventions and some exceptional cases such as COVID-19 studies must receive prior approval from the University's medical ethics committee. However, surveys conducted among persons who are not patients and that are limited to answering an anonymous questionnaire do not require approval by the medical ethics committee. Anyway, the surveys were conducted in strict compliance with research standards, including voluntary participation and respect for privacy and confidentiality for each participant and was carried out in compliance with Congolese law with approval of the Ministry of Public Health through the Programme National de Lutte Contre les Toxicomanies et les Substances Toxiques [National Program for the Fight against Drug Addiction and Toxic Substances (PNLCT)].

A brief introduction of the questionnaire is necessary.

Response: A paragraph was added in the methods section to describe briefly the questionnaire.

Results

Lines 154-155, that could be avoided at the beginning of the study if the participant information sheet has clearly stated the criteria of participation.

Response: Thank you for this observation, in the methodology we included the following paragraph: “Sometimes in the region, it is unusual, for privacy reasons, to disclose one’s age. Some old pupils prefer to keep their age secret because they are ashamed; only teachers or head masters have access to the file of pupils, and they are not allowed to disclose it without pupils consent. Some older pupils participated to the survey just to prove to their colleagues that they have normal school age. To respect privacy, the ages were at our disposal at the end of the survey”.

Other comments

Please read the paper carefully with regard to correct English

Response:  Thank you for this comment. We have made a number of corrections.

Reviewer 3 Report

The review is attached.

Author Response

REPLY TO COMMENTS

Reviewer 3

In the opinion of the reviewer, the work contains too many errors to be published.

Response: we respectfully disagree with the reviewer; admittedly, our work has limitations but it does not contain “many errors”

Introduction:

This section should be reworded thoroughly. It should justify the purpose of the work and focus, among others, on the health aspects of energy drink consumption, differences in the drinks sold (composition, price). Please justify why such a research group was selected. What are the earnings in the Democratic Republic of Congo? How much do energy drinks cost? How high is the consumption of energy drinks in the DRC? What does alcohol consumption look like? Is It Common For Children To Drink Alcohol? Are there legal regulations regarding the consumption of energy drinks? Do they concern adults, children? Is there an obligation to provide the nutritional value on the labels of food products? What must be on the labels? Who is responsible for the legal regulations regarding food products, including energy drinks.

Response: To our opinion, the purpose of the study is well justified. The issues raised by the reviewer were thoroughly addressed in the manuscript, either in the results (see table 1) or in the discussion. 

What are the earnings in the Democratic Republic of Congo? How high is the consumption of energy drinks in the DRC?

Response: The DRC is a low income country [GDP per capita: US$581 in 2019 according to the World Bank (https://data.worldbank.org/indicator/NY.GDP.PCAP.CD?locations=CD)] (information added to revised manuscript) and we clearly said in the manuscript that our study is the first investigation of ED consumption among adolescent pupils in the DRC. We did not find a study on the general consumption of energy drinks in DRC, the only one found was a publication conducted among university student in the same area and we cited it in our study.

What does alcohol consumption look like? Is It Common For Children To Drink Alcohol?

Response: the legal age for alcohol consumption in the DRC is 18 years, but according to data from the WHO
http://www.who.int › publications › profiles › cod PDF , the prevalence of heavy drinking among 15-19 y olds is 19%.  In our study, we focused on the consumption of energy drinks and we also assessed the frequency of mixing energy drinks with alcohol..

Is there an obligation to provide the nutritional value on the labels of food products? What must be on the labels? Who is responsible for the legal regulations regarding food products, including energy drinks.

Response: In our study, we did not address the issue of nutritional value of energy drinks. It is the responsibility of the company producers to provide nutritional information and with warnings for potential dangerous substances, and of the government to control the legislation application. The International Council of Beverages Associations (ICBA) made Guidelines for the Composition, Labelling and Responsible Marketing of Energy Drinks. Explanations can be found in the following references cited in our study:

  • Health Canada. Category Specific Guidance for Temporary Marketing Authorization - Caffeinated Energy Drinks. Health Canada. December 2013. http://www.hc-sc.gc.ca/fn-an/alt_formats/pdf/legislation/guide-ld/guidance-caf-drink-boiss-tma-amt-eng.pdf
  • International Council of Beverages Associations (ICBA). Guidelines for the Composition, Labelling and Responsible Marketing of Energy Drinks 2013. https://www.icba-net.org/files/resources/energy-drink-guidelines.pdf

But each country can adapt the international legislation to the local reality to protect the population.

Line 54. Please add a literature reference (has considerably increased worldwide [ ])

Response: There is a literature reference at 54, “[1 ]”.

Lines: 59-61. Please add a literature reference to this passage.

Response: The reference to this part was added

Lines: 64-65. Please rewrite. The excerpt is not understandable.

Response: We do not understand why the excerpt is not understandable on the line 64-65.

Lines: 69-73. Please delete or link this passage to the authors' work.

Response: Passage of Lines 69-73 are linked to 4 authors’ work in the manuscript, we do not understand why we have to delete it or to link authors’ work.

Materials and methods.

How many people took part in the survey? What was the division according to age and gender?

Response: We think that responses of these questions are clearly defined in the two first paragraphs of the Results section.

Line: 149 What are the proportions?

Response: The proportions (prevalences) of participants according to their responses.

What was the survey questionnaire like? What parts was it made of?

Response: A paragraph was added in the methods section to describe briefly the questionnaire.

How were the drinks selected for the research? Why 12 kinds?

Response: we simply bought the available brands in super markets.

Line: 143 How many determinations were made?

Why was it decided to label the caffeine content if it was on the labels?

Response: It was necessary to check if the information on the label was correct.

Results

Lines: 152-167. This fragment should be included in the Materials and Methods section.

Response: Thank you for this suggestion but as this information is a “result” we think that it should be important to put it at the beginning of the results section.

Table 2.

Please explain why the authors chose to formulate the question (Reason for not consuming ED: "Ban by parents or by teachers"). How much influence do teachers in the Democratic Republic 76 of Congo have on children? Is it so big that they can forbid children, for example, from consuming selected products?

Response: The influence of teachers is apparent from our study results. Teachers can represent parents, and can support adolescent for public health or security concern. A recent study conducted in Sweden among adolescents was added in the discussion section and can also provide more details about this issue.

Discussion

The excerpts in this section duplicate those in the Results section.

Response: based on good publication practice, we first summarized the salient results of the survey. We also expanded the discussion as suggested by another reviewer.:

  • Svensson, Å.; Warne, M.; Gillander Gådin,. K. Longitudinal Associations Between Energy Drink Consumption, Health, and Norm-Breaking Behavior Among Swedish Adolescents. Front. Public Health 2021, 9:597613. doi: 10.3389/fpubh.2021.597613
  • Visram, S; Crossley, SJ; Cheetham, M; Lake, A. Children and young people's perceptions of energy drinks: A qualitative study. PLoS ONE 2017, 12(11): e0188668. doi.org/10.1371/journal.pone.0188668

Conclusions are generalized.

Response: Thank you for pointing the conclusion, but we think that our conclusions are reasonable.

Round 2

Reviewer 2 Report

The manuscript has been improved.

For proposed research involving human participants, either involves direct contact with participants or does not, ethical approval is needed. There might be different regulations in different countries, so it will be the decision of editorial office.

Why did you measure caffeine concentrations since caffeine concentrations have been indicated on the labels? please explain it.

Author Response

REPLY TO COMMENTS

Reviewer 2

The manuscript has been improved.

Response: thanks for this positive comment

For proposed research involving human participants, either involves direct contact with participants or does not, ethical approval is needed. There might be different regulations in different countries, so it will be the decision of editorial office.

Response: as we explained in the methodology section and in our previous reply, surveys conducted among persons who are not patients and that are limited to answering an anonymous questionnaire do not at our institution require approval by the medical ethics committee.

For your information, the study on energy drinks consumption by university students done in the same area and cited in our study [22], did not address ethical aspects. Another study from our institution also proves this:

Mulungulungu, N.H.A.D.; Mukakakera, S. Dépenses énergétiques et consommations alimentaires des transporteurs à vélo du charbon de bois à Lubumbashi/Energetical expenses and food consumptions of charcoal bicycle carriers in Lubumbashi. International Journal of innovation and Applied Studies 2017, 19(4),813-823. http://www.proquest.com/docview/18542211585/fulltextPDF/758173F9AE2D44DEPQ/1?accountid=17215

Why did you measure caffeine concentrations since caffeine concentrations have been indicated on the labels? Please explain it.

Response: As explained in the manuscript, we wanted to check if the information on the label was correct. As shown by our study and by other publications cited in our article [34-37], the content indicated on the labels does not always truly reflect the actual content. This is why government laboratories in many countries (though not in the DRC, as far as we know) routinely test the composition of commercial food and drinks.

Reviewer 3 Report

The authors generally did not address the comments of the reviewer.

They made only minor corrections in work, which concerned supplementing the content with literature references. On lines 79-80, they entered the income per capita figures for 2019 in the Democratic Republic of Congo. However, they did not associate this with the purpose of the work.

Therefore, the reviewer maintains that the work in its current form contains significant errors and should be redrafted. 

Author Response

REPLY TO COMMENTS

Reviewer 3

The authors generally did not address the comments of the reviewer.

They made only minor corrections in work, which concerned supplementing the content with literature references. On lines 79-80, they entered the income per capita figures for 2019 in the Democratic Republic of Congo. However, they did not associate this with the purpose of the work.

Therefore, the reviewer maintains that the work in its current form contains significant errors and should be redrafted.

Response:  we tried to address the criticisms of the reviewer but we do not understand why the reviewer still considers that our study contains "significant errors" because neither he/she, nor the other two reviewers pointed to errors in the design, conduct, analysis or interpretation of our study; we, therefore, leave it to the editor to decide whether our article is suitable for publication in IJERPH.